# LLM-Informed Semi-Supervised Learning for Text Classification

## Abstract

Large Language Models (LLMs) have shown impressive zero-shot and few-shot capabilities in many NLP tasks including text classification. While these models outperform others in terms of raw performance when few examples are available, they are expensive to use in practice and may lag behind traditional approaches when labeled (or unlabeled) data is plentiful. Semi-supervised learning (SSL) can utilize large amounts of unlabeled data in combination with labeled data to improve a model's performance. In this paper, we propose to unify LLM and SSL under a common framework which effectively leverages the few-shot capabilities of LLMs in combination with SSL's ability to extract valuable information from unlabeled data to improve model performance in text classification tasks. Our approach, called LLM-SSL, utilizes LLMs to generate predictions on unlabeled examples and uses these predictions to guide the SSL training and improve the quality of pseudo-labels during training. We show that LLM-SSL outperforms both prior SSL models as well as few-shot LLMs on six text classification benchmarks.

## 1 Introduction

Semi-supervised learning (SSL) has emerged as a powerful approach with good few-shot learning capabilities. SSL mitigates the requirement for large labeled datasets by using a model itself to assign pseudo-labels to unlabeled data, and thus, effectively makes use of information from unlabeled data Chen et al. (2020b); Arazo et al. (2020); Xie et al. (2020a). To ensure high-quality pseudo-labels, SSL approaches Lee et al. (2013); Xie et al. (2020a); Sohn et al. (2020) leverage fixed high-confidence thresolds during training, which allow the model to access unlabeled examples during training only if the model's confidence on these examples is very high.

While methods that employ high-confidence thresholds such as FixMatch Sohn et al. (2020) are shown to consistently reduce the confirmation bias Arazo et al. (2019), these rigid thresholds allow access only to a small amount of unlabeled data for training, and thus, ignore a considerable amount of unlabeled (and diverse) examples for which the model's predictions do not exceed the fixed confidence threshold. Zhang et al. (2021) introduced FlexMatch that relaxes the rigid confidence threshold to account for the model's learning status of each class and adaptively scales down the threshold for a class to encourage the model to learn from more examples from that class. Moreover, Chen et al. (2023) proposed SoftMatch that does not discard any unlabeled examples—no matter how low their confidence is, but assigns adaptive weights according to the model's confidence. Both FlexMatch and SoftMatch have access to a much larger and diverse set of unlabeled data to learn from, but lowering or eliminating the thresholds can lead to the introduction of wrong pseudo-labels (even if with a low weight), which are extremely harmful for generalization.

To address this drawback and mitigate the harmful effects of wrong pseudo-labels, we leverage Large Language Models (LLMs) Wei et al. (2022); Kojima et al. (2022a); Gao et al. (2021); Schick & Schütze (2021), a line of research which has produced models with great zero-shot and few-shot capabilities and combine it with SSL. Our approach entitled LLM-SSL consists of two separate models: an LLM that uses few-shot prompting and a BERT model that is fine-tuned on labeled and pseudo-labeled data over multiple training iterations. We utilize the LLM to generate predictions on unlabeled examples and to guide the BERT model especially in the early stages of training when BERT is not very robust but rather produces unreliable or noisy predictions (due to small supervised data used for training). Specifically, we distill the knowledge of LLM through a novel *teaching*

*annealing* method that fuses the pseudo-labels generated by BERT during SSL training with those generated by the LLM, assigning a higher weight to the LLM predictions if the BERT model has a poor learning status (as measured on the training set) and gradually lowering the weight of the LLM predictions if the BERT model has a high learning status. In addition, LLM-SSL reduces the amount of incorrect pseudo-labels during training by analyzing the behavior of the model on unlabeled data from the start of training up until the end instead of relying solely on the confidence of the model at a single iteration to impose the confidence threshold Sohn et al. (2020); Zhang et al. (2021); Xie et al. (2020a). We estimate the correctness of a pseudo-label using margins Bartlett et al. (2017); Pleiss et al. (2020); Elsayed et al. (2018); Jiang et al. (2018); Sosea & Caragea (2023) of unlabeled examples averaged across the training iterations. We believe that this type of thresholding that takes into account the entire training history is more expressive and can encode more information about a pseudo-label, leading to better data quality. Our approach is lightweight, requiring significantly less compute than LLMs in practice and incurring smaller computational cost during training compared to other SSL methods (since it starts with higher quality pseudo-labels provided by LLMs and gradually switches to BERT pseudo-labels as the BERT becomes increasingly more robust).

We carry out comprehensive experiments using various experimental setups on six SSL text classification benchmarks: IMDB Maas et al. (2011), RCV1 Lewis et al. (2004), GoEmotions Demszky et al. (2020), Amazon Review McAuley & Leskovec (2013), TREC-6 Li & Roth (2002), and Yelp Review Asghar (2016). We find that LLM-SSL yields significant improvements on all benchmarks, outperforming strong LLM and SSL baselines. Notably, our method outperforms FlexMatch and SoftMatch by $7.5\%$ and $5.34\%$, respectively, in accuracy on IMDB using 20 labels per class and by as much as $7.3\%$ and $2.8\%$ on Amazon Review using 20 labels per class.

Our contributions are as follows: **1)** We combine SSL and LLM under a common framework: LLM-SSL allows access to a large set of unlabeled data to learn from and enforces high pseudo-label quality during training by leveraging the vast knowledge of LLMs and by monitoring the training dynamics of unlabeled data as training progresses to detect and filter out potentially incorrect pseudo-labels; **2)** We show that LLM-SSL outperforms existing works on six well-established text classification benchmarks showing larger improvements in error rates especially on challenging datasets, while achieving similar convergence performance (or better) than strong prior works; **3)** We perform a comprehensive analysis of our approach and indicate potential insights into why our LLM-SSL substantially outperforms other techniques.

## 2 RELATED WORK

We first discuss related work on semi-supervised learning for text classification. Second, we discuss Large Language Models (LLMs) and their application to text classification.

**Semi-supervised Learning** Semi-supervised learning has attracted much attention in the NLP community Gururangan et al. (2019); Yang et al. (2015); Clark et al. (2018); Chen et al. (2020a); Yang et al. (2017); Chen et al. (2020a); Xie et al. (2020a); Mukherjee & Awadallah (2020); Miyato et al. (2016); Wang et al. (2022); Yang et al. (2023); Shi et al. (2023); Huang et al. (2023); Tan et al. (2024), since unlabeled data is often much easier to acquire compared to labeled data. Yang et al. (2019) used a hierarchy structure to propagate supervision from high-level labels to lower-level labels, while Clark et al. (2018) introduced cross-view training, where a model makes auxiliary predictions only seeing parts of the input text and is trained to match the predictions when given the entire input. Mukherjee & Awadallah (2020) introduced uncertainty estimates into self-training (UST), a particular type of SSL where a teacher and a student are iteratively trained using labeled and unlabeled data. SoftMatch Chen et al. (2023) does not utilize any threshold and instead dynamically weights each unlabeled example during training, assigning lower weights to examples whose pseudo-labels are potentially incorrect and higher weights otherwise. MarginMatch Sosea & Caragea (2023) monitors the training dynamics of unlabeled examples during SSL training and at an arbitrary iteration imposes a threshold that takes into account the behavior of the model from the beginning of training up until the current iteration. We use UST, SoftMatch, and MarginMatch as strong baselines in our experiments.

Consistency regularization Sajjadi et al. (2016) is an important component in recent semi-supervised learning approaches and relies on the continuity assumption Bachman et al. (2014); Laine & Aila (2017) that the model should output similar predictions on multiple perturbed versions of the same input example. Popular approaches such as Unsupervised Data Augmentation (UDA) Xie

et al. (2020a), FixMatch Sohn et al. (2020) and FlexMatch Zhang et al. (2021) use consistency regularization at their core combined with psedo-labeling. In psedo-labeling Lee et al. (2013), a model itself is used to assign artificial labels for unlabeled data and only artificial labels whose largest class probability is above a predefined confidence threshold are used during training. We consider UDA, FixMatch and FlexMatch as strong baselines, and compare LLM-SSL against them in all our experiments. While these SSL methods maintain pseudo-label quality using high-confidence thresholds, in fully supervised learning, Area-Under-the-Margin (AUM) Pleiss et al. (2020) is a popular technique to ensure high quality labels by monitoring the training dynamics of examples and removing potentially mislabeled examples from training. Inspired by this work, we leverage training dynamics of unlabeled examples to maintain qualitative pseudo-labels during training.

**Large Language Models** Language models (LMs) are models that estimate the probability distribution over text. Recently, improvements through larger amounts of data (e.g. WebText Gao et al. (2020)) and increasingly larger model sizes (from a few million Merity et al. (2016) to hundreds of millions Devlin et al. (2019) to hundreds of billions Brown et al. (2020) parameters) have enabled pre-trained large language models (LLMs) to be incredibly powerful at solving many downstream NLP tasks. In the past, language models were used in a *pre-train and fine-tune* manner where a language model is pretrained on a large unlabeled corpus then adapted to a target task by fine-tuning Devlin et al. (2019). However, it was recently observed that scaling models to 100B+ parameters leads to capabilities of few-shot learning Brown et al. (2020) by way of in-context learning. One can guide the model generation by simply designing a prompt to solve the task, starting an era of *pre-train and prompt* Liu et al. (2023). In this work, we use one of these proposed chain-of-thought prompting (CoT) Kojima et al. (2022b) techniques for few-shot predictions. We subsequently use these predictions in our SSL framework to substantially improve SSL performance.

## 3 LLM-SSL

**Notation** Let $L = \{(x_1, y_1), ..., (x_B, y_B)\}$ be a batch of size $B$ of labeled examples and $U = \{\hat{x}_1, ..., \hat{x}_{\nu B}\}$ be a batch of size $\nu B$ of unlabeled examples, where $\nu$ is the batch-wise ratio of unlabeled to labeled examples. Let $p_\theta(y|x)$ denote the class distribution produced by model $\theta$ on example $x$ and $\hat{p}_\theta(y|x)$ denote the argmax of this distribution as a one-hot label. Let $\pi(x)$ and $\Pi(x)$ denote a weak and strong augmentation of an input example $x$. Additionally, let $H(p, q)$ be the cross-entropy between two probability distributions $p$ and $q$.

### 3.1 BACKGROUND

We build and improve upon FlexMatch Zhang et al. (2021). FlexMatch argued that using a high *fixed* threshold $\tau$ (as in FixMatch Sohn et al. (2020)) to filter the (potentially erroneous) pseudo-labeled data ignores the learning difficulties of different classes and prevents the model from seeing diverse and challenging unlabeled examples (i.e., examples that do not pass the confidence threshold) which, if used properly, may improve the capabilities of the model. To this end, FlexMatch adaptively scales the confidence threshold $\tau$ depending on the learning status of each class, assuming that a class with fewer examples above the fixed threshold $\tau$ has a greater learning difficulty, and hence, it adaptively lowers the threshold $\tau$ to encourage more training examples from this class to be learned. The learning status $\alpha_c$ for a class $c$ is simply computed as the number of unlabeled examples that are predicted in class $c$ and pass the fixed threshold $\tau$:

$$\alpha_c = \sum_{i=1}^n \mathbb{1}(\max(p_\theta(y|\pi(\hat{x}_i))) > \tau) \times \mathbb{1}(\hat{p}_\theta(y|\pi(\hat{x}_i)) = c) \tag{1}$$

where $n$ is the total number of unlabeled examples. This learning effect is then normalized and used to obtain the class-dependent threshold for each class $c$:

$$\mathcal{T}_c = \frac{\alpha_c}{\max_c(\alpha_c)} \times \tau \tag{2}$$

In practice, FlexMatch iteratively computes new thresholds after each complete pass through unlabeled data, hence we can parameterize $\mathcal{T}_c$ as $\mathcal{T}_c^t$, denoting the threshold obtained at iteration $t$.

## 3.2 PROPOSED APPROACH: LLM-SSL

While the flexible thresholds of FlexMatch allow access to a diverse set of unlabeled data, these thresholds can introduce incorrect pseudo-labels, which are harmful due to confirmation bias Lee et al. (2013). To address this drawback, we propose to analyze the historical predictions of the model on unlabeled examples instead of relying solely on the confidence at an iteration. At the same time, since the model predictions at the beginning of training are usually not reliable, we propose to leverage the world knowledge of LLMs through a novel teacher annealing framework to guide the SSL training process during the early stages of training.

### 3.2.1 MONITORING TRAINING DYNAMICS OF UNLABLED EXAMPLES

We use the margin of a training example Bartlett et al. (2017); Pleiss et al. (2020); Elsayed et al. (2018); Jiang et al. (2018) to estimate the quality of the example as being correctly annotated or potentially mislabeled. The margin quantifies the difference between the logit corresponding to the assigned ground truth label and the largest other logit. In our SSL formulation, since no ground truth is available for unlabeled data we define the margins as *pseudo-margins*. Let $c$ be the pseudo-label (or the argmax of the prediction, i.e., $\hat{p}_\theta(y|\pi(\hat{x}))$) at iteration $t$ on unlabeled example $\hat{x}$ after applying weak augmentations. We define the *pseudo-margin* (PM) of $\hat{x}$ with respect to pseudo-label $c$ at iteration $t$ as follows:

$$\text{PM}_c^t(\hat{x}) = z_c - max_{i!=c}(z_i) \tag{3}$$

where $z_c$ is the logit corresponding to the assigned pseudo-label $c$ and $\max_{i!=c}(z_i)$ is the largest *other* logit corresponding to a label $i$ different from $c$. To monitor the model's predictions on $\hat{x}$ with respect to pseudo-label $c$ from the beginning of training to iteration $t$, we average all the margins with respect to $c$ from the first iteration until $t$ and obtain the average pseudo-margin (APM) as follows:

$$\text{APM}_c^t(\hat{x}) = \frac{1}{t}\sum_{j=1}^{t}\text{PM}_c^j(\hat{x}) \tag{4}$$

Here $c$ acts as the "ground truth" label for the APM calculation. Note that if at a prior iteration $t'$, the assigned pseudo-label is different from $c$ (say $c'$), then the APM calculation at iteration $t'$ is done with respect to $c'$ (by averaging all margins with respect to $c'$ from 1 to $t'$). In practice, we maintain a vector of pseudo-margins for all classes accumulated over the training iterations and dynamically retrieve the accumulated pseudo-margin value of the argmax class $c$ to obtain the $\text{APM}_c^t$ at iteration $t$.

Intuitively, if $c$ is the pseudo-label of $\hat{x}$ at iteration $t$, then $\text{PM}_c^t$ with respect to class $c$ at iteration $t$ will be positive. In contrast, if the argmax of the model prediction on $\hat{x}$ at a previous iteration $t' < t$ is different from $c$, then $PM_c^{t'}$ at $t'$ with respect to $c$ will be negative. Therefore, if over the iterations, the model predictions do not agree frequently with the pseudo-label $c$ from iteration $t$ and the model fluctuates significantly between iterations on the predicted label, the APM for class $c$ will have a low, likely negative value. Similarly, if the model is highly uncertain of the class of $\hat{x}$ (reflected in a high entropy of the class probability distribution), the APM for class $c$ will have a low value. These capture the characteristics of mislabeled examples or of those harmful for training. Motivated by these observations, LLM-SSL leverages the APM of the assigned pseudo-label $c$ and compares it with an APM threshold $\gamma$ to mask out pseudo-labeled examples with low APMs.

**Exponential Moving Average of Pseudo-Margins** The current definition of APM weighs the pseudo-margin at iteration $t$ identical to the pseudo-margin at a much earlier iteration $p$ ($p << t$). This is problematic since very old pseudo-margins eventually become deprecated (especially due to the large number of iterations through unlabeled data in consistency training ($\sim 9K$)), and hence, the old margins are no longer indicative of the current learning status of the model. To this end, instead of averaging all pseudo-margins (from the beginning of training to the current iteration), we propose to use an exponential moving average to place more importance on recent iterations. Formally, APM becomes:

$$\text{APM}_c^t(\hat{x}) = \text{PM}_c^t(\hat{x}) * \frac{\delta}{1+t} + \text{APM}_c^{t-1}(\hat{x}) * (1 - \frac{\delta}{1+t}) \tag{5}$$

---

**Algorithm 1** LLM-SSL

---

**Require:** Labeled data $L$; unlabeled data $U$; maximum number of iterations $T$; number of classes $C$; $\theta$ base model; $\theta^{LLM}$ LLM model, $\pi$ weak augmentations; $\Pi$ strong augmentations, APM threshold $\gamma$.

1: Generate hard (one-hot) pseudo-labels for the every unlabeled example $\hat{x}_i \in U$ using $\theta^{LLM}$:

$$\hat{y}^{LLM}(\hat{x}_i) = \hat{p}_{\theta^{LLM}}(\hat{x}_i)$$

2: **for** $t$ = 1 to $T$ **do**
3:     Estimate the learning status $\alpha_c$ (Eq. 1) of model $\theta$ and calculate the class-wise flexible thresholds $\mathcal{T}_c^t$     (Eq. 2) for each class $c$.
4:     **while** $U$ not exhausted **do**
5:         Labeled batch $L_b = \{(x_1, y_1), ..., (x_B, y_B)\}$, unlabeled batch $U_b = \{\hat{x}_1, ..., \hat{x}_{\nu B}\}$
6:         **for** $x \in U_b$ **do**
7:             Compute logits $z_c$ using model $\theta$ for each class $c$ after applying weak augmentations.
8:             Calculate pseudo-margin $PM_c^t$ (Eq. 3) and update Average $PM_c^t$ (Eq. 4) for each class $c$.
9:         **end for**
10:     Compute normalized class-wise learning status:

$$S_c = \frac{\alpha_c}{\max_c(\alpha_c)}$$

11:     Compute pseudo-labels by fusing the LLM ($\theta^{LLM}$) and base model ($\theta$) predictions:

$$\hat{y}(\hat{x}_i) = S_{\hat{p}_\theta(y|\pi(\hat{x}_i))} \times \hat{p}_\theta(y|\pi(\hat{x}_i)) + (1 - S_{\hat{p}_\theta(y|\pi(\hat{x}_i))}) \times \hat{y}^{LLM}(\hat{x}_i)$$

12:     Minimize $\mathcal{L} = \mathcal{L}_s + \lambda \mathcal{L}_u$ where

$$\mathcal{L}_s = \frac{1}{B} \sum_{i=1}^{B} H(y_i, p_\theta(y|\pi(x_i)))$$

$$\mathcal{L}_u = \sum_{i=1}^{\nu B} \mathbb{1}(\text{APM}_{\hat{p}_\theta(y|\pi(\hat{x}_i))}^t(\hat{x}_i) > \gamma) \times \mathbb{1}(\max(p_\theta(y|\pi(\hat{x}_i))) > \mathcal{T}_{\hat{p}_\theta(y|\pi(\hat{x}_i))}^t) \times H(\hat{y}(\hat{x}_i), p_\theta(y|\Pi(\hat{x}_i)))$$

13:     **end while**
14: **end for**

---

We set the smoothing parameter $\delta$ to 0.997 in experiments.

### 3.2.2 LLM FOR PSEUDO-LABEL GENERATION

During the early stages of training when the learning status of the model is poor, incorrect pseudo-labels have a higher chance of being propagated to the next iterations since the generalization capabilities of the model are limited. We propose to use powerful general-knowledge models such as LLMs to improve the quality of pseudo-labeled data especially at the beginning of training. Specifically, we enhance model training using a novel teacher annealing framework that mixes the predictions of the LLM with those of our SSL model: in the early stages when the SSL model is weak we rely more on the predictions of the LLM and as the SSL model becomes more robust, we gradually increase the weight of the prediction of the SSL model.

Due to its superior performance and efficiency, we use an instruction-tuned Mistral Jiang et al. (2023) $7B$ parameter model (Mistral-7B-Instruct) with few-shot chain-of-thought (CoT) prompting to generate pseudo-labels on all the unlabeled examples:

$$\hat{y}^{LLM}(\hat{x}_i) = \hat{p}_{\theta^{LLM}}(\hat{x}_i) \tag{6}$$

As we mentioned previously, we assign higher importance to these pseudo-labels when the learning status of the model is poor. Specifically, we leverage the class-dependent normalized learning status from FlexMatch (Eq. 2):

$$S_c = \frac{\alpha_c}{\max_c(\alpha_c)} \tag{7}$$

We use this learning status to merge the model-generated and LLM-generated pseudo-labels. We assign a greater weight to the model-generated pseudo-label (lower weight to the LLM) if the model learning status is high, and a lower weight otherwise (higher weight to the LLM):

$$\hat{y}(\hat{x}_i) = S_{\hat{p}_\theta(y|\pi(\hat{x}_i))} \times \hat{p}_\theta(y|\pi(\hat{x}_i)) + (1 - S_{\hat{p}_\theta(y|\pi(\hat{x}_i))}) \times \hat{y}^{LLM}(\hat{x}_i) \tag{8}$$

Note that the resulting pseudo-label $\hat{y}(\hat{x}_i)$ is not necessarily a one-hot label. Specifically, if the LLM and our model's predictions differ, the pseudo-label will be two-hot. Formally, the unlabeled loss in LLM-SSL becomes:

$$\mathcal{L}_u = \sum_{i=1}^{\nu B} \mathbb{1}(\text{APM}_{\hat{p}_\theta(y|\pi(\hat{x}_i))}^t(\hat{x}_i) > \gamma) \times \mathbb{1}(\max(p_\theta(y|\pi(\hat{x}_i))) > \mathcal{T}_{\hat{p}_\theta(y|\pi(\hat{x}_i))}^t) \times H(\hat{y}(\hat{x}_i), p_\theta(y|\Pi(\hat{x}_i)))$$
(9)

where $\gamma$ is the APM threshold and $\mathcal{T}_{\hat{p}_\theta(y|\pi(\hat{x}_i))}^t$ is the flexible threshold estimated as in FlexMatch Zhang et al. (2021). To train our model, we adopt the best practices Zhang et al. (2021); Sohn et al. (2020) and optimize the weighted combination of the supervised and unsupervised losses:

$$\mathcal{L} = \mathcal{L}_s + \lambda \mathcal{L}_u$$
(10)

where the supervised loss is given by:

$$\mathcal{L}_s = \frac{1}{B} \sum_{i=1}^{B} H(y_i, p_\theta(y|\pi(x_i)))$$
(11)

Our full LLM-SSL algorithm is shown in Algorithm 1.

## 4 EXPERIMENTS AND RESULTS

In this section, we first introduce the six benchmark text classification datasets that we used to evaluate LLM-SSL (§4.1). Next, we present the baselines used for comparison with our LLM-SSL (§4.2). Finally, we detail our experimental setup (§4.3) and discuss the results that we obtain on all datasets in low supervised data regimes (§4.4).

### 4.1 DATASETS

We experiment with the following benchmark datasets to asses the effectiveness of our method: (1) **IMDB** Maas et al. (2011) is a movie review dataset annotated at review level with the positive and negative labels; (2) **RCV1** Lewis et al. (2004) is a large scale benchmark dataset composed of news stories labeled with a total of 105 different topics; (3) **GoEmotions** Demszky et al. (2020) is a sentence-level emotion detection dataset created using Reddit comments. GoEmotions is annotated with 27 emotions and the neutral class, and provides a great opportunity to study the expression of fine-grained emotions and to develop emotion classification models; (5) **Amazon Review** McAuley & Leskovec (2013) is a sentiment classification dataset of Amazon reviews annotated with 5 sentiment classes. (5) **TREC-6** Li & Roth (2002) is a dataset where fact-based questions are divided into six broad semantic categories; (6) **Yelp Review** Asghar (2016) is a sentiment classification dataset composed of Yelp reviews annotated with 5 sentiment classes.

### 4.2 BASELINES

First, we carry out experiments using fully supervised and LLM-based models. **BASE** is a fully supervised approach obtained by training a BERT Devlin et al. (2019) base uncased model on the labeled data only. **LLM-ZS** (LLM-Zero Shot) obtains predictions using our LLM in a zero-shot fashion. **LLM-FS** (LLM-Few Shot) obtains predictions using our LLM using few-shot CoT prompting. This is the model used to obtain predictions in our LLM-SSL framework. **LLM-DS** (LLM-Distant Supervision) uses the LLM to generate labels on all the unlabeled examples. Then it trains a BERT model in a supervised fashion on the union of labeled data and newly LLM-generated pseudo-labeled data. **LLM-DS-AUM** is very similar to LLM-DS, however, during training on the union of labeled data and pseudo-labeled data we also monitor the AUM of the pseudo-labeled examples. Then, similar to Pleiss et al. (2020), we remove the low-AUM pseudo-labeled examples and train the model again.

Second, we carry out experiments using semi-supervised approaches based on a teacher-student framework. **NOISY-S** (Noisy Student Training) Xie et al. (2020b) involves generating pseudo-labels for unlabeled data and iteratively training models in a teacher-student setup. **UST** (Uncertainty-aware Self-

Training) Mukherjee & Awadallah (2020) incorporates uncertainty estimates into the standard self-training framework by adding a few highly effective changes. UST computes uncertainty estimates for all unlabeled examples by stochastically passing the examples from this set through the model multiple times, with dropout enabled before each layer. The approach subsequently uses these uncertainty estimates to select what unlabeled data to use. Concretely, the model not only favors unlabeled data where the teacher model is confident, but also enforces low entropy of the teacher predictions.

Third, we experiment with approaches based on consistency regularization. **UDA** (Unsupervised Data Augmentation) Xie et al. (2020a) leverages Backtranslation Edunov et al. (2018) and uses a consistency loss to enforce the model predictions on unlabeled data to be invariant to input noise. **FixMatch** Sohn et al. (2020) predicts artificial labels for unlabeled examples using a weakly-augmented version of each unlabeled example and then employs the artificial labels as pseudo-labels to train against but this time using a strongly-augmented version of each unlabeled example. FixMatch uses unlabeled examples solely if the confidence of the model prediction exceeds a *fixed* threshold. **FlexMatch** Zhang et al. (2021) argued that prior methods ignore the learning difficulties of different classes, and introduced class-dependent thresholds to account for these learning difficulties. We presented FlexMatch in detail in Section §3.1. **SoftMatch** Chen et al. (2023) eliminates the thresholds completely and instead dynamically weights the unlabeled examples during training, assigning lower weights to potentially mislabeled examples and higher weight otherwise. **MarginMatch** Sosea & Caragea (2023) utilizes training dynamics of unlabeled examples to design a more effective threshold for eliminating incorrect pseudo-labels.

## 4.3 EXPERIMENTAL SETUP

We evaluate the performance of our LLM-SSL by varying the number of training examples on the six text classification benchmark datasets presented above. On each dataset, we experiment with 20, 50, 100, and 200 labeled examples per class, which we sample without replacement. The remaining examples are used as unlabeled data. We follow the exact evaluation metrics used in the works introducing the datasets: accuracy for IMDB, TREC-6, Amazon Review, Yelp Review and macro F1 for GoEmotions and RCV1. In each setup, we also run our models three times, with different parameter intializations, and report the average results, as well as their standard deviations. All our experiments use BERT Devlin et al. (2019) base uncased as the backbone model which is trained for 200 epochs. We use the translation models provided by Tiedemann & Thottingal (2020) for backtranslation. In terms of augmentations, we create weakly augmented examples using synonym replacement Kolomiyets et al. (2011) and SwitchOut Wang et al. (2018) by randomly performing one or both augmentations. In terms of strong augmentations, we perform a random combination of Back-translations Tiedemann & Thottingal (2020) using long chain lengths ($> 5$), SwitchOut and synonym replacements. In all our experiments, the baselines use the same weak and strong augmentations as LLM-SSL. For few-shot prompts we leverage a maximum of 5 examples per class in the prompt with a value lower than 5 if the prompt exceeds the maximum Mistral context window sequence length (4096 tokens). The few-shot examples originate from the small available labeled set (e.g., the set of 20/50/100/200 examples per class). We emphasize that our trained model is easy to use in practice, requiring only inference using our BERT model. While our method involves using an LLM during training, we also note that we only run inference once on the unlabeled set (Step 1 in Algorithm 1).

## 4.4 RESULTS

We show the results obtained across the six datasets in Table 1. First, we note that our zero-shot LLM-ZS obtains very good results and outperforms our fully supervised model significantly on datasets such as IMDB and RCV1. Notably, using 20 examples per class on IMDB, LLM-ZS ourperforms BASE by a considerable 12.0% accuracy. Additionally, our LLM-DS approach is very competitive as well outperforming SSL methods in some setups. For instance, using 50 labels per class on RCV1, LLM-DS outperforms FixMatch by a significant 1.2% in accuracy.

We also emphasize that among our SSL baselines, approaches based on consistency learning are the most effective. Specifically, FlexMatch Zhang et al. (2021) and SoftMatch Chen et al. (2023) are the second best performing models in most of the results. For example, SoftMatch considerably outperforms both Noisy student and UST in accuracy on GoEmotions using 20 labels per class by 10% and 8%, respectively.

| Dataset (Metric) | IMDB (Accuracy) | | | | RCV1 (F1) | | | | GoEmotions (F1) | | | |
|---|---|---|---|---|---|---|---|---|---|---|---|---|
| Num Labels | 20 lb/cl | 50 lb/cl | 100 lb/cl | 200 lb/cl | 20 lb/cl | 50 lb/cl | 100 lb/cl | 200 lb/cl | 20 lb/cl | 50 lb/cl | 100 lb/cl | 200 lb/cl |
| BASE | $69.1_{4.25}$ | $78.4_{+3.55}$ | $75.3_{+2.65}$ | $80.3_{+2.35}$ | $22.1_{+4.55}$ | $24.19_{+4.36}$ | $26.88_{+3.65}$ | $27.2_{+3.56}$ | $09.2_{+6.34}$ | $20.4_{+4.51}$ | $26.3_{+2.35}$ | $21.4_{+2.15}$ |
| LLM-ZS | 81.1 | 81.1 | 81.1 | 81.1 | 53.5 | 53.5 | 53.5 | 53.5 | 19.3 | 19.3 | 19.3 | 19.3 |
| LLM-FS | 84.2 | 84.2 | 84.2 | 84.2 | 57.9 | 57.9 | 57.9 | 57.9 | 23.1 | 23.1 | 23.1 | 23.1 |
| LLM-DS | $80.0_{4.12}$ | $82.4_{+3.65}$ | $83.1_{+3.24}$ | $84.5_{+2.55}$ | $55.1_{+4.21}$ | $57.5_{+3.56}$ | $58.1_{+3.24}$ | $62.0_{+3.07}$ | $24.1_{+2.86}$ | $24.7_{+2.65}$ | $24.1_{+2.55}$ | $27.6_{+2.31}$ |
| LLM-DS-AUM | $81.4_{3.11}$ | $82.2_{+1.25}$ | $83.4_{+1.45}$ | $85.1_{+1.65}$ | $56.3_{+2.18}$ | $58.1_{+2.11}$ | $59.0_{+2.31}$ | $62.4_{+1.76}$ | $25.3_{+4.12}$ | $25.7_{+4.04}$ | $24.6_{+3.78}$ | $28.1_{+3.21}$ |
| NOISY-S | $71.1_{2.41}$ | $81.3_{+2.35}$ | $76.5_{+2.25}$ | $81.5_{+2.18}$ | $42.3_{+3.56}$ | $43.22_{+3.41}$ | $20.45_{+3.17}$ | $45.1_{+3.41}$ | $15.4_{+5.18}$ | $25.3_{+3.51}$ | $24.1_{+3.21}$ | $21.9_{+2.51}$ |
| UST | $72.5_{2.71}$ | $82.1_{+2.51}$ | $74.9_{+2.36}$ | $82.7_{+2.21}$ | $52.3_{+4.25}$ | $54.1_{+3.68}$ | $42.7_{+3.32}$ | $53.1_{+2.51}$ | $17.4_{+2.41}$ | $23.1_{+2.23}$ | $24.6_{+2.16}$ | $\underline{31.5}_{+2.31}$ |
| UDA | $77.4_{2.45}$ | $83.15_{+2.22}$ | $79.7_{+2.01}$ | $83.7_{+2.07}$ | $54_{+4.51}$ | $56.7_{+4.21}$ | $59.3_{+3.58}$ | $62.1_{+3.21}$ | $24.5_{+3.05}$ | $25.9_{+5.12}$ | $26.4_{+4.51}$ | $30.9_{+4.14}$ |
| FixMatch | $81.5_{2.61}$ | $84.2_{+2.32}$ | $88.5_{+2.14}$ | $86.5_{+2.08}$ | $54.4_{+3.51}$ | $56.3_{+3.25}$ | $60.2_{+3.11}$ | $62.4_{+2.38}$ | $23.9_{+4.63}$ | $24.9_{+4.21}$ | $25.8_{+3.87}$ | $30.0_{+3.51}$ |
| FlexMatch | $80.4_{3.96}$ | $86.3_{+3.52}$ | $\underline{89.7}_{+3.34}$ | $90.2_{+3.31}$ | $56.5_{+4.56}$ | $58.9_{+4.21}$ | $\underline{61.7}_{+3.95}$ | $64.1_{+3.76}$ | $24.7_{+4.61}$ | $\underline{26.2}_{+4.51}$ | $26.1_{+4.21}$ | $30.3_{+3.56}$ |
| SoftMatch | $\underline{82.56}_{2.31}$ | $\underline{88.12}_{+2.15}$ | $89.5_{+1.02}$ | $\underline{91.5}_{+1.75}$ | $\underline{58.6}_{+3.56}$ | $\underline{60.2}_{+3.41}$ | $61.2_{+3.25}$ | $\underline{65.2}_{+3.05}$ | $\underline{25.4}_{+4.31}$ | $26.1_{+4.25}$ | $\underline{26.5}_{+3.31}$ | $30.7_{+3.41}$ |
| MarginMatch | $82.5_{3.31}$ | $87.5_{+3.21}$ | $86.1_{+4.22}$ | $86.8_{+2.15}$ | $56.3_{+1.87}$ | $58.4_{+2.11}$ | $62.2_{+1.52}$ | $64.1_{+1.42}$ | $23.9_{+2.34}$ | $27.1_{+2.51}$ | $29.5_{+2.15}$ | $31.2_{+2.22}$ |
| LLM-SSL | $\mathbf{87.9_{2.08}}$ | $\mathbf{89.5_{+2.31}}$ | $\mathbf{91.9_{+1.98}}$ | $\mathbf{92.3_{+2.21}}$ | $\mathbf{61.5_{+3.56}}$ | $\mathbf{62.9_{+3.01}}$ | $\mathbf{64.9_{+2.52}}$ | $\mathbf{66.4_{+2.11}}$ | $\mathbf{28.2_{+3.14}}$ | $\mathbf{27.4_{+3.65}}$ | $\mathbf{27.9_{+2.76}}$ | $\mathbf{32.4_{+2.45}}$ |

| Dataset (Metric) | TREC-6 (Accuracy) | | | | Amazon Review (Accuracy) | | | | Yelp Review (Accuracy) | | | |
|---|---|---|---|---|---|---|---|---|---|---|---|---|
| Num Labels | 20 lb/cl | 50 lb/cl | 100 lb/cl | 200 lb/cl | 20 lb/cl | 50 lb/cl | 100 lb/cl | 200 lb/cl | 20 lb/cl | 50 lb/cl | 100 lb/cl | 200 lb/cl |
| BASE | $80.4_{2.51}$ | $85.1_{+2.31}$ | $83.3_{+2.05}$ | $85.9_{+1.67}$ | $47.6_{+3.15}$ | $48.4_{+3.01}$ | $53.1_{+2.44}$ | $54.2_{+2.31}$ | $38.5_{+1.45}$ | $46.2_{+1.52}$ | $49.2_{+1.23}$ | $58.7_{+0.92}$ |
| LLM-ZS | 75.2 | 75.2 | 75.2 | 75.2 | 44.7 | 44.7 | 44.7 | 44.7 | 44.7 | 44.7 | 44.7 | 44.7 |
| LLM-FS | 82.4 | 82.4 | 82.4 | 82.4 | 49.1 | 49.1 | 49.1 | 49.1 | 48.8 | 48.8 | 48.8 | 48.8 |
| LLM-DS | $75.1_{1.86}$ | $78.1_{+1.96}$ | $82.7_{+1.75}$ | $87.4_{+1.86}$ | $46.9_{+3.42}$ | $51.3_{+2.45}$ | $57.6_{+1.37}$ | $59.1_{+2.17}$ | $46.3_{+1.45}$ | $48.4_{+1.53}$ | $52.7_{+2.31}$ | $57.5_{+1.76}$ |
| LLM-DS-AUM | $76.8_{2.18}$ | $78.6_{+2.07}$ | $83.4_{+1.89}$ | $88.0_{+1.67}$ | $47.2_{+1.78}$ | $53.6_{+2.15}$ | $58.1_{+1.97}$ | $59.0_{+1.56}$ | $48.1_{+1.53}$ | $49.2_{+2.33}$ | $54.2_{+1.45}$ | $59.2_{+1.46}$ |
| NOISY-S | $81.3_{2.42}$ | $86.2_{+2.21}$ | $89.2_{+2.02}$ | $84.5_{+2.11}$ | $46.4_{+3.12}$ | $50.2_{+2.85}$ | $56.3_{+1.33}$ | $58.7_{+2.07}$ | $46.5_{+1.55}$ | $47.3_{+1.53}$ | $51.6_{+2.23}$ | $56.9_{+1.45}$ |
| UST | $82.2_{1.54}$ | $87.5_{+1.94}$ | $89.3_{+1.888}$ | $85.9_{+2.21}$ | $45.2_{+2.36}$ | $51.5_{+1.78}$ | $57.9_{+1.41}$ | $58.3_{+2.07}$ | $46.8_{+1.31}$ | $49.1_{+1.73}$ | $52.3_{+2.51}$ | $56.8_{+1.55}$ |
| UDA | $84.2_{2.31}$ | $88.1_{+2.15}$ | $89.7_{+1.97}$ | $\mathbf{91.6_{+1.87}}$ | $47.1_{+3.31}$ | $52.1_{+1.21}$ | $56.5_{+1.32}$ | $58.7_{+2.01}$ | $45.8_{+1.24}$ | $49.4_{+1.21}$ | $53.5_{+1.99}$ | $57.8_{+1.53}$ |
| FixMatch | $86.7_{1.98}$ | $89.1_{+1.91}$ | $89.1_{+1.95}$ | $91.0_{+1.88}$ | $46.0_{+2.35}$ | $51.1_{+2.22}$ | $56.8_{+1.54}$ | $58.7_{+2.41}$ | $47.2_{+1.26}$ | $48.1_{+1.48}$ | $52.5_{+1.56}$ | $53.8_{+1.11}$ |
| FlexMatch | $88.4_{4.22}$ | $88.4_{+3.58}$ | $89.2_{+3.22}$ | $90.7_{+1.56}$ | $46.9_{+3.42}$ | $51.3_{+2.45}$ | $57.6_{+1.37}$ | $59.1_{+2.17}$ | $46.3_{+1.45}$ | $48.4_{+1.53}$ | $52.7_{+2.31}$ | $57.5_{+1.76}$ |
| SoftMatch | $\underline{89.6}_{2.21}$ | $\underline{90.1}_{+2.11}$ | $\underline{90.1}_{+1.65}$ | $90.0_{+1.71}$ | $51.4_{+1.43}$ | $53.7_{+2.44}$ | $58.9_{+2.41}$ | $59.3_{+1.44}$ | $51.6_{+2.06}$ | $55.9_{+1.78}$ | $\underline{58.55}_{+1.54}$ | $60.1_{+1.67}$ |
| MarginMatch | $87.6_{3.56}$ | $88.9_{+2.41}$ | $89.8_{+2.33}$ | $90.4_{+2.55}$ | $52.3_{+1.67}$ | $52.7_{+1.75}$ | $57.8_{+1.56}$ | $59.3_{+1.44}$ | $51.7_{+1.52}$ | $56.4_{+1.58}$ | $58.3_{+2.13}$ | $60.1_{+2.54}$ |
| LLM-SSL | $\mathbf{90.3_{2.15}}$ | $\mathbf{90.9_{+1.65}}$ | $\mathbf{90.4_{+1.55}}$ | $91.5_{+1.51}$ | $\mathbf{54.2_{+1.45}}$ | $\mathbf{56.4_{+1.23}}$ | $\mathbf{60.8_{+1.45}}$ | $\mathbf{61.1_{+1.42}}$ | $\mathbf{52.1_{+1.51}}$ | $\mathbf{57.9_{+1.64}}$ | $\mathbf{60.8_{+1.57}}$ | $\mathbf{62.7_{+1.41}}$ |

Table 1: Results on six text classification benchmarks in various low data regime setups. The best model is colored and the second best model is underlined.

Overall, we observe that the proposed LLM-SSL is extremely effective, significantly outperforming strong baselines on all datasets. For example, LLM-SSL pushes the accuracy over UDA by an average of $7.6\%$ on IMDB and $6.4\%$ on RCV1. Critically, using only 100 examples per class on Yelp Review, LLM-SSL obtains $60.8\%$ accuracy, an improvement of $2.2\%$ over the second best performing SSL method SoftMatch. Moreover, compared to UDA and the fully supervised BERT, we see an improvement of around $7.3\%$ and $11.6\%$ respectively. We also see consistent improvements on GoEmotions, where our method significantly improves performance in all setups. For instance, we improve upon the fully supervised baseline BASE model by $19\%$ in F1 score using 20 examples per class and by $11\%$ using 200 examples per class.

# 5 ANALYSIS

## 5.1 ABLATION STUDY

We perform an ablation study to tease apart the components that lead to the success of our LLM-SSL. To this end, we design the following variations of LLM-SSL and train them in all settings (20/50/100/200 labels per class on the six datasets): **1) APM** method discards completely the LLM component of our approach. In this version, the model uses solely its predictions on weakly augmented examples to generate the pseudo-labels. **2) LLM-SSL**$^{naive}$ is similar to our main LLM-SSL but instead of using the learning status of the model to weigh the LLM and base model pseudo-labels (i.e., the $S$ term in Equation 8) it weighs them equally instead (i.e., $S = 0.5$). **3) LLM-SSL**$^{noAPM}$ eliminates the APM-based threshold from Equation 9, keeping only the confidence threshold. **4) LLM-SSL**$^{agreement}$ is similar to **LLM-SSL**$^{naive}$ but instead of weighing the pseudo-labels of the base model and the LLM equally, our model only uses the pseudo-label if the base model and the LLM agree (i.e., the LLM and base model pseudo-labels are the same).

We show the results obtained in Table 2. Discarding the LLM completely (i.e., APM method) leads to considerable degradations on all datasets. Notably, we see a degradation in accuracy of $5.4\%$ on IMDB using 20 examples per class. Along the same lines, on RCV1 we also observe a degradation of $5.2\%$ in the same setup. Interestingly, removing the APM metric from our approach (LLM-SSL$^{noAPM}$) also decreases the performance, indicating that the combination of LLM with the APM-based threshold both contribute to the observed benefits. At the same time, using the LLM to impose an additional threshold (LLM-SSL$^{agreement}$) significantly lowers the performance of our method. For example, our LLM-SSL outperforms LLM-SSL$^{agreement}$ by $3.0\%$ F1 on GoEmotions using 20 examples per class and by $4.2\%$ on IMDB in the same setup. Finally, we observe that our LLM-SSL outperforms

| Dataset (Metric) | IMDB (Accuracy) | | | | RCV1 (F1) | | | | GoEmotions (F1) | | | |
|---|---|---|---|---|---|---|---|---|---|---|---|---|
| Num Labels | 20 lb/cl | 50 lb/cl | 100 lb/cl | 200 lb/cl | 20 lb/cl | 50 lb/cl | 100 lb/cl | 200 lb/cl | 20 lb/cl | 50 lb/cl | 100 lb/cl | 200 lb/cl |
| APM | 82.5 | 87.5 | 86.1 | 86.8 | 56.3 | 58.4 | 62.2 | 64.1 | 23.9 | 27.1 | 29.5 | 31.2 |
| LLM-SSL$^{naive}$ | 84.2 | 88.1 | 87.4 | 92.0 | 58.9 | 59.5 | 62.3 | 64.9 | 25.7 | 26.1 | 27.3 | 31.2 |
| LLM-SSL$^{agreement}$ | 83.7 | 87.3 | 90.2 | 90.4 | 51.0 | 58.9 | 61.2 | 64.1 | 25.2 | 26.5 | 27.1 | 31.2 |
| LLM-SSL$^{noAPM}$ | 86.7 | 89.2 | 91.1 | 91.2 | 58.4 | 61.5 | 64.1 | 65.4 | 26.5 | 26.9 | 27.7 | 29.8 |
| LLM-SSL | 87.9 | 89.5 | 91.9 | 92.3 | 61.5 | 62.9 | 64.9 | 66.4 | 28.2 | 27.4 | 27.9 | 32.4 |
| Dataset (Metric) | TREC-6 (Accuracy) | | | | AMAZON REVIEW (Accuracy) | | | | YELP REVIEW (Accuracy) | | | |
| Num Labels | 20 lb/cl | 50 lb/cl | 100 lb/cl | 200 lb/cl | 20 lb/cl | 50 lb/cl | 100 lb/cl | 200 lb/cl | 20 lb/cl | 50 lb/cl | 100 lb/cl | 200 lb/cl |
| APM | 87.6 | 88.9 | 89.8 | 90.4 | 50.2 | 53.4 | 57.5 | 59.5 | 50.3 | 54.4 | 59.6 | 58.7 |
| LLM-SSL$^{naive}$ | 87.1 | 90.1 | 90.2 | 91.2 | 52.3 | 55.4 | 59.4 | 58.4 | 50.4 | 55.6 | 58.7 | 60.3 |
| LLM-SSL$^{agreement}$ | 88.7 | 89.0 | 88.7 | 90.4 | 53.1 | 55.2 | 59.9 | 60.3 | 50.5 | 55.4 | 58.1 | 60.5 |
| LLM-SSL$^{noAPM}$ | 89.1 | 89.1 | 89.3 | 90.1 | 53.2 | 54.9 | 58.7 | 60.7 | 50.4 | 55.1 | 59.3 | 61.2 |
| LLM-SSL | 90.3 | 90.9 | 90.4 | 91.5 | 54.2 | 56.4 | 60.8 | 61.1 | 52.1 | 57.9 | 60.8 | 62.7 |

Table 2: Ablation study of our method. The best model is colored and the second best model is underlined.

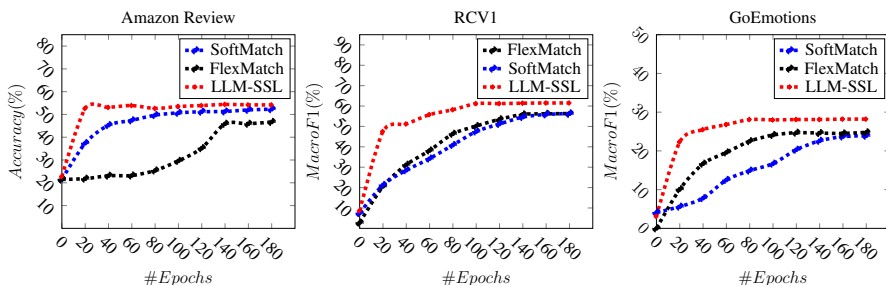

Figure 1: Convergence speed of LLM-SSL as compared with that of FlexMatch and SoftMatch on three datasets using 20 labels per class.

LLM-SSL$^{naive}$ significantly as well by an average of $2.3\%$ on all the datasets. All these results showcase the effectiveness of LLM-SSL and that all its components contribute to its success.

## 5.2 CONVERGENCE SPEED

We plot in Figure 1 the performance of LLM-SSL against SoftMatch and FlexMatch during the entire training process on three datasets with varying number of classes in low-resource settings to understand the convergence performance of our LLM-SSL approach. We observe common trends across the experiments: LLM-SSL achieves high performance much quicker compared to both SoftMatch and FlexMatch, indicating that the LLM predictions boost the capabilities of the model to learn quickly in the early stages of training. Notably, on Amazon Review LLM-SSL attains $52\%$ accuracy after only 20 epochs whereas SoftMatch obtains the same performance at the 100th epoch. Similarly, on both RCV1 and GoEmotions the performance of LLM-SSL rises at a significantly faster pace, indicating that it converges quicker compared to FlexMatch and SoftMatch.

## 6 CONCLUSION

We improve semi-supervised learning in text classification by **1)** Introducing a novel Average Pseudo-Margin unlabeled example selection technique and **2)** Leveraging LLMs through a novel teacher annealing framework to incorporate external knowledge into our model. We show that our approach is effective in a wide range of domains (social networks, forums, online platforms) and contexts (movie reviews, medical forum discussions) and outperforms other strong SSL approaches. In the future, we plan to study our method in settings where there is a mismatch between the labeled and unlabeled data distributions and analyze how we can use out-of-domain unlabeled data to boost the performance.

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
