# OpenReview forum: "LLM-Informed Semi-Supervised Learning for Text Classification"
_ICLR.cc/2025/Conference — Submitted to ICLR 2025_

### Official Review · Reviewer_hxCX · 2024-10-19

**Soundness:** 3
**Presentation:** 2
**Contribution:** 2
**Rating:** 5
**Confidence:** 4

**Summary:**

Summary:

This work proposes a method to make an LLM to generate pseudo labels in a semi-supervised way to train another models on text-classification tasks.

Contribution:

The experimental results of this method on text-classification tasks are good compared to some settings such as the few-shot in-context learning of LLMs.

**Strengths:**

The experimental results of this method on text-classification tasks are good compared to some settings, such as the few-shot in-context learning of LLMs.

**Weaknesses:**

1. The cost of this work can be quite large compared to the non-training method of utilizing unlabeled texts by language models.

2. The backbone is not good enough to capture complex semantics of different downstream tasks which is likely to fail in the text-generation tasks.

3. The tasks are quite simple which involves largely emotion classification.

4. There are many advanced few-shot methods for in-context learning. However, the author has not mentioned them.

5. The captions of tables are not up to the standard of the ICLR submission format.

**Questions:**

1. I wonder about the results of advocating more advanced few-shot methods, such as better retrieval or chain-of-thought methods. I am unsure whether we need to train an encoder-only model to solve the emotion classification task.

---

> ### Author Response · Authors · 2024-11-28
>
> > The cost of this work can be quite large compared to the non-training method of utilizing unlabeled texts by language models.
>
> One of our motivations of the paper is distilling the information of LLMs into a small model such as BERT. So  our method needs significantly less compute. We make this point clear in L64-L68 and also indicate that a major advantage of the method is having to perform inference using LLM only once on the unlabeled data. Please note that Mistral-7B can yield 1 request / 200ms on a single A100. This means only ~3hours are needed to pseudo-label the unlabeled data in the GoEmotions setup (45K unlabeled examples). In contrast our BERT model can perform 1 request / 1ms on the same A100, a 200X speedup. That means during deployment our approach is 200X less intensive than leveraging the LLM and delivers higher performance (as shown in our experiments). We will add a discussion on this point.
>
>
> > The backbone is not good enough to capture complex semantics of different downstream tasks which is likely to fail in the text-generation tasks.
>
> We would like to ask for more clarifications to this point. We chose BERT due to its compactness and ease of use in practice. We deal with classification tasks so the text-generation capabilities are not relevant to our paper.
>
> > The tasks are quite simple which involves largely emotion classification.
>
> We would like to point out to the reviewer that we did not perform experiments largely on emotion classification. This comment seems to be completely unrelated to our paper. Only 1 dataset out of the 6 is an emotion dataset.
>
> > There are many advanced few-shot methods for in-context learning. However, the author has not mentioned them.
>
> Our goal in the paper was not to obtain the most optimal LLM setup, which can be domain and task specific. Instead, our goal was to show that leveraging a good-enough LLM in a teacher-annealing SSL framework improves both the performance of the LLM and that of the SSL methods. We do expect that better LLM models yield better LLM-SSL results. In fact, comparing LLM-SSL-FS to LLM-SSL-ZS is evidence towards this, where the few-shot LLM applied in our method yields better performance than the zero-shot version. We would also like to remind you that one of the core strengths of our method is its computational efficiency, since the inference needs to be performed solely using the BERT model.
>
> > The captions of tables are not up to the standard of the ICLR submission format.
>
> We would like more details on this point. The comment is very generic. What is not “up to the standard of the ICLR submission format”?

---

### Official Review · Reviewer_k6WR · 2024-11-03

**Soundness:** 2
**Presentation:** 2
**Contribution:** 2
**Rating:** 5
**Confidence:** 4

**Summary:**

The paper porposed LLM-SSL, a novel approach that combines Large Language Models (LLMs) with Semi-Supervised Learning (SSL) for text classification tasks. The primary goal is to improve the quality of pseudo-labels generated during SSL by leveraging the vast knowledge of LLMs, particularly in the early stages of training when the model's predictions are less reliable. Additionally, the approach introduces Average Pseudo-Margin (APM) metric to monitor the training dynamics of unlabeled examples, ensuring high-quality pseudo-labels throughout training. Experiments on six text classification benchmarks show significant improvements over strong baselines.

**Strengths:**

1. The proposed method (including APM and LLM) is well-motivated and novel.

1. The experimental results seems very competitive.

**Weaknesses:**

1. Some details in experiments in unclear to me, e.g., what's weak augmentation and strong augmentation?
1. I did not see discussion about APM threshold $\gamma$. In my understanding, it should be a important hyperparam to control our confidence of pseudo labels. I want to see more evidence to show its robustness.

**Questions:**

Please refer to weaknesses.

---

> ### Author Response · Authors · 2024-11-28
>
> > Some details in experiments in unclear to me, e.g., what's weak augmentation and strong augmentation?
>
> We mention in L356-L359  that we use synonym replacement and switch-out as weak augmentations and a random combination of Backtranslations [Tiedemann & Thottingal (2020)] using long chain lengths (> 5), SwitchOut and synonym replacements as strong augmentation.
>
> > I did not see discussion about APM threshold. In my understanding, it should be a important hyperparam to control our confidence of pseudo labels. I want to see more evidence to show its robustness.
>
> Thank you for the comment. We set the AUM threshold as the AUM of the 95th percentile (i.e., we eliminate 5% of the examples with the lowest AUMs). We will make this clear in the paper.

---

### Official Review · Reviewer_t5CC · 2024-11-03

**Soundness:** 3
**Presentation:** 3
**Contribution:** 2
**Rating:** 5
**Confidence:** 4

**Summary:**

The paper presents LLM-SSL, a novel approach that combines semi-supervised learning (SSL) with few-shot prompting from large language models (LLMs) for text classification. In this method, the LLM generates pseudo-labels for unlabeled data, while SSL techniques, relying on training dynamics, leverage these pseudo-labels to enhance model training. Specifically, the authors propose using a class-specific exponential moving average of pseudo-margins over training iterations to estimate the model's certainty in each pseudo-label and refine these labels by combining predictions from both the LLM and the model.

Experiments were conducted on six datasets across 20- to 200-shot cases, comparing several baselines. These experiments showed consistent improvements over the baselines. An additional ablation study further explores the contributions of each component within the method.

**Strengths:**

1. Overall, the paper is well-written and clear, though minor edits are needed.
2. The motivation is well-established.
3. The proposed method is novel, building on established SSL literature, including works like FixMatch, FlexMatch, SoftMatch, and MarginMatch.
4. Results demonstrate consistent improvements over the baselines.
5. The method shows faster convergence.
6. The ablation study reveals that the different components are needed for achieving the overall performance.

**Weaknesses:**

1. The paper does not compare itself to several key methods in the field of few-shot text classification, which would provide important context. These include:
A. Basic few-shot text classification methods, such as SetFit and FastFit.
B. Self- and semi-supervised methods specifically designed for classification. For instance, the paper "Zero-Shot Text Classification with Self-Training" achieves strong results on datasets like IMDB and GoEmo in a zero-shot setup, although by using unlabeled training data without augmentation. Several other methods in this category could also be relevant for comparison.
C. LLM-based approaches for enhancing classification. For example, "Selective In-Context Data Augmentation for Intent Detection using Pointwise V-Information" leverages LLM-generated examples and employs smaller classifiers training dynamic to to determine whether to include these examples in training. "In-Context Learning for Text Classification with Many Labels" uses retrieval models to select the most relevant examples for LLM few-shot contexts.

Including comparisons or at least discussing key distinctions between LLM-SSL and these methods is essential for demonstrating the relevance and advantages of the proposed approach in few-shot text classification.

2. Although much of the method presentation focuses on the SSL aspect, ablation results show that LLM-SSL without the APM component still performs well, closely matching LLM-SSL’s overall performance and outperforming most baselines. This raises questions about the specific impact of the SSL component, as much of the improvement may stem from the integration of the LLM with FlexMatch, rather than the SSL itself.

3. Regarding the experimental setup, only Mistral is used as the LLM throughout the experiments. Given that an essential part of the method involves the LLM, it would be valuable to assess its impact by experimenting with different LLMs. This would clarify the LLM’s role within the method, which is currently underexplored.

4. Although less critical, the use of only one encoder model, such as BERT (which is relatively dated), is a limitation. Including newer models like RoBERTa, DeBERTa, or other recent alternatives could demonstrate the robustness of the method across various base architectures.

**Questions:**

1. Why LLM ZS and FS are the same across all shots (in all cases the shots exceed the context!?) and don't have STD results?
2. The SSL extension of the method can be very general (not necessarily for languages, and not for classification) and can target many different tasks. How will it perform on other tasks?
3. In the pseudo-code you describe the algorithm as changing the data, is it not easier to describe it by changing the data?

Comments:

1. No explanation on the way the STD was calculated, no STD for ablation, no significant testing.

2. Small points:

a. row 167: grammar, confidence at an iteration

b. row 182: \neq can be a better annotation in eq. 3

c. row 219:1 in the pseudo-code:  pseudo-labels "for the every unlabeled example" -> for every

d. pseudo-labels line 8, avreage PM, why not APM, why eq. 4 and not 5?

e. pseudo-labels line 12, the loss function H is not defined.

f. row 253: superior to what? which specific model version?

g. you mention that the pseudo-label can be a two-hot vector but didn't explain how it affects the loss function.

h. row 331: why n Edunov is not in brackets, same 333, verify for all the paper.

Probably will be good to proofread the whole paper.

---

> ### Author Response · Authors · 2024-11-28
>
> > The paper does not compare itself to several key methods in the field of few-shot text classification, which would provide important context. These include: A. Basic few-shot text classification methods, such as SetFit and FastFit. B. Self- and semi-supervised methods specifically designed for classification. For instance, the paper "Zero-Shot Text Classification with Self-Training" achieves strong results on datasets like IMDB and GoEmo in a zero-shot setup, although by using unlabeled training data without augmentation. Several other methods in this category could also be relevant for comparison. C. LLM-based approaches for enhancing classification. For example, "Selective In-Context Data Augmentation for Intent Detection using Pointwise V-Information" leverages LLM-generated examples and employs smaller classifiers training dynamic to to determine whether to include these examples in training. "In-Context Learning for Text Classification with Many Labels" uses retrieval models to select the most relevant examples for LLM few-shot contexts. Including comparisons or at least discussing key distinctions between LLM-SSL and these methods is essential for demonstrating the relevance and advantages of the proposed approach in few-shot text classification.
>
> Thank you for your comment. Our goal in the paper was not to obtain the most optimal LLM setup, which can be domain and task specific. Instead, our goal was to show that leveraging a good-enough LLM in a teacher-annealing SSL framework improves both the performance of the LLM and that of the SSL methods. We do expect that better LLM models yield better LLM-SSL results. In fact, comparing LLM-SSL-FS to LLM-SSL-ZS is evidence towards this, where the few-shot LLM applied in our method yields better performance than the zero-shot version. We would also like to remind you that one of the core strengths of our method is its computational efficiency, since the inference needs to be performed solely using the BERT model.
>
> Thank you for your baselines SSL recommendations. Our goal was to compare our approach with SSL works that are similar (consistency regularization-based) and contemporary to our work. For completeness, we will add the suggested comparisons.
>
>
> > Although much of the method presentation focuses on the SSL aspect, ablation results show that LLM-SSL without the APM component still performs well, closely matching LLM-SSL’s overall performance and outperforming most baselines. This raises questions about the specific impact of the SSL component, as much of the improvement may stem from the integration of the LLM with FlexMatch, rather than the SSL itself.
>
> Thank you for your comment. We showed in our ablation experiment in Table 2 that both the APM and the LLM pseudo-labels lead to performance improvements. Concretely, as can be seen from the Table, the LLM-SSL^{no_amp} method, which removed the APM thresholding obtains an average of $1.1\%$ less performance compared to LLM-SSL. This result indicates that the APM component boosts the performance of our method.
>
> > Regarding the experimental setup, only Mistral is used as the LLM throughout the experiments. Given that an essential part of the method involves the LLM, it would be valuable to assess its impact by experimenting with different LLMs. This would clarify the LLM’s role within the method, which is currently underexplored.
>
> Thank you for your comment. We will include results using open-source methods like LLama-3, as well as closed source models (GPT-4o) in the paper. We have previously carried out LLama-2 experiments and have observed similar trends. See the Table below with results for 20 labels/class:
>
>
> |                   | IMDB | RCV1 | GoEmotions | TREC-6 | Amazon Review | Yelp Review |
> | ----------------- | ---- | ---- | ---------- | ------ | ------------- | ----------- |
> | LLM-SSL (Mistral) | 87.9 | 61.5 | 28.2       | 90.3   | 54.2          | 52.1        |
> | LLM-SSL (LLama 2) | 87.3 | 61.2 | 27.5       | 89.4   | 53.7          | 52.1        |
>
>
>
> > Although less critical, the use of only one encoder model, such as BERT (which is relatively dated), is a limitation. Including newer models like RoBERTa, DeBERTa, or other recent alternatives could demonstrate the robustness of the method across various base architectures.
>
> We will add an analysis with different encoder models. Thank you for your suggestion.

---

> > ### Comment · Reviewer_t5CC · 2024-11-28
> >
> > Thank you for your response.
> > 1. You state that your aim is "not to obtain the most optimal LLM setup, which can be domain- and task-specific," yet you apply your method to a specific task. In my view, you could either apply your method to a broader range of tasks to demonstrate its generality or compare it to other classification baselines. Regarding computational efficiency, other classification baselines are also efficient.
> > 2. I agree that APM improved your method. However, my point was that the improvement from APM is relatively small compared to the improvement contributed by the LLM.
> > 3. While adding a few LLM experiments is an important first step, they are not sufficient to "clarify the LLM’s role within the method, which is currently underexplored."
> > 4. Like in #3, the main point is to better understand the role of the encoder in the method.
> >
> > Overall, I think the paper has the potential to make a significant contribution. However, framing the method primarily as a classification approach may limit its perceived generality. Applying the method to additional tasks, such as NLI, sentiment analysis, or similar problems, could better demonstrate its robustness. Furthermore, exploring the use of multi-modal LLMs and extending the method to vision tasks could enhance its versatility. Finally, a deeper analysis of the interplay between the trained model (e.g., BERT) and the LLM (e.g., Mistral) would, in my view, significantly improve the overall quality of the paper.

---

> > > ### Author Response · Authors · 2024-12-02
> > >
> > > Thank you for engaging with us in the discussion!
> > >
> > > > You state that your aim is "not to obtain the most optimal LLM setup, which can be domain- and task-specific," yet you apply your method to a specific task. In my view, you could either apply your method to a broader range of tasks to demonstrate its generality or compare it to other classification baselines. Applying the method to additional tasks, such as NLI, sentiment analysis, or similar problems, could better demonstrate its robustness.
> > >
> > > Thank you for the comment. We underline that our paper focuses on semi-supervised learning for text. Similar to prior works in SSL [1], we focus solely on classification. Please note that we cover a wide range of classification tasks, ranging from sentiment analysis (IMDB) and emotion detection (GoEmotions) to question classification (TREC-6) and topic classification (RCV1). We will look into expanding to additional domains.
> > >
> > >
> > > [1] - USB: A Unified Semi-supervised Learning Benchmark for Classification
> > >
> > >
> > > > While adding a few LLM experiments is an important first step, they are not sufficient to "clarify the LLM’s role within the method, which is currently underexplored. " Like in #3, the main point is to better understand the role of the encoder in the method.
> > >
> > >
> > > We remind here what the LLM’s role is in our approach: during the early stages of training when the learning status of the model is poor, incorrect pseudo labels have a higher chance of being propagated to the next iterations since the generalization capabilities of the SSL model are limited. Therefore we propose to use powerful general-knowledge models such as LLMs to improve the quality of pseudo-labeled data especially at the beginning of training.
> > >
> > >
> > > > I agree that APM improved your method. However, my point was that the improvement from APM is relatively small compared to the improvement contributed by the LLM.
> > >
> > > The analysis presented in the ablation in Table 2 is meant to indicate the effectiveness of the two proposed core components in the paper: the APM thresholding and our pseudo-label fusion approach. We notice that in the reviews the second component is largely overlooked and seen as “just using LLM labels in a framework”. Please note that we do not just use the LLM labels as-is. In fact, **LLM-SSL-naive which does this, considerably lags behind LLM-SSL**.
> > >
> > > Regarding your point, yes, LLM predictions do improve more since these help the SSL model start from a better learning status, which decreases the confirmation bias that stems from incorrect predictions in early training. But we would like to underline that this improvement stems from **our label fusion contribution** and should be treated as such.
> > >
> > > Additionally, APM does complement our goal of preventing confirmation bias; after all, this is its main goal. The performance improvements obtained from APM are also not insignificant; there are improvements across the board, most times more than $1$% with improvements of up to $3.1$% on RCV1.
> > >
> > > We believe our contributions and what this paper tries to achieve is a bit misunderstood. The fact that the weak labels come from the LLM is not even that important. Our method shows that utilizing some signal (in this case LLM labels) to carefully complement the SSL model in early training in combination with stricter pseudo-label quality approaches (i.e., APM) leads to considerable gains on SSL benchmarks.
> > >
> > > We will make sure to make this point clear in the paper.

---

### Official Review · Reviewer_MfPv · 2024-11-04

**Soundness:** 2
**Presentation:** 3
**Contribution:** 2
**Rating:** 1
**Confidence:** 5

**Summary:**

This paper studies the semi-supervised text classification problem with the help of LLMs. The method is built upon FlexMatch (Zhang et al., 2021) and further introduces hard pseudo-labels from LLMs for the unlabeled texts. The LLM's pseudo-labels and traditional pseudo-labels from FlexMatch are blended together during the iterations and then used as the supervision for the model training. The experiment results on 6 datasets show that the proposed method outperforms both FlexMatch and LLM, showing that the integration of both methods is effective.

**Strengths:**

- This paper works on a well-defined problem by introducing LLM as help.
- The proposed method is intuitive and easy-to-follow.
- Six datasets are comprehensive.

**Weaknesses:**

- Integrating LLM into the semi-supervised text classification problem is a natural idea, however, the way proposed in this paper is a little straightforward. Based on the line 1 in Algorithm 1, we will have to send all the unlabeled texts to the LLM, which can be very expensive. We can of course expect the performance improvement as the classifier is trained with additional signals from LLM, however, perhaps at a huge cost. An analysis between cost and improvement will be very useful. I hope to see a cost-effective method to improve the classification.

- Many weakly supervised text classification works have been ignored in this paper. Their problem setting often involves very limited supervised data, and sometimes, even zero annotated documents. The final classification performance can be very strong too (e.g., 90% F1 or higher). For example, some recent works [1, 2] in this direction also discussed how to leverage language models to better generate pseudo-labels and put them into a bootstrap framework for classifier training. More recently, label selection and debiasing have been studied too [3]. I believe some of these methods can be easily adapted as baselines here.

[1] Mekala, Dheeraj, and Jingbo Shang. "Contextualized weak supervision for text classification." Proceedings of the 58th Annual Meeting of the Association for Computational Linguistics. 2020.

[2] Wang, Zihan, Dheeraj Mekala, and Jingbo Shang. "X-Class: Text Classification with Extremely Weak Supervision." Proceedings of the 2021 Conference of the North American Chapter of the Association for Computational Linguistics: Human Language Technologies. 2021.

[3] Dong, Chengyu, Zihan Wang, and Jingbo Shang. "Debiasing Made State-of-the-art: Revisiting the Simple Seed-based Weak Supervision for Text Classification." Proceedings of the 2023 Conference on Empirical Methods in Natural Language Processing. 2023.

**Questions:**

- This is a minor format issue. I think the authors somehow didn't know how to use the citation properly in this ICLR template. Please explore \citet and \citep and check their differences. In many cases, I believe you may want to use (NAME et al., YEAR) instead of NAME et al., (YEAR).

---

> ### Author Response · Authors · 2024-11-28
>
> > Integrating LLM into the semi-supervised text classification problem is a natural idea, however, the way proposed in this paper is a little straightforward. Based on the line 1 in Algorithm 1, we will have to send all the unlabeled texts to the LLM, which can be very expensive. We can of course expect the performance improvement as the classifier is trained with additional signals from LLM, however, perhaps at a huge cost. An analysis between cost and improvement will be very useful. I hope to see a cost-effective method to improve the classification.
>
> One of our motivations of the paper is distilling the information of LLMs into a small model such as BERT. So our method needs significantly less compute. We make this point in L64-L68 and also indicate that a major advantage of the method is having to perform inference using the LLM only once on the unlabeled data. Please note that Mistral-7B can yield 1 request / 200ms on a single A100. This means only \~3hours are needed to pseudo-label the unlabeled data in the GoEmotions setup (\~45K examples). In contrast our BERT model can perform 1 request / 1ms on the same A100, a 200X speedup. That means during deployment our approach is 200X less intensive than leveraging the LLM and delivers higher performance (as shown in our experiments). We will add a discussion on this point.
>
>
> > Many weakly supervised text classification works have been ignored in this paper. Their problem setting often involves very limited supervised data, and sometimes, even zero annotated documents. The final classification performance can be very strong too (e.g., 90% F1 or higher). For example, some recent works [1, 2] in this direction also discussed how to leverage language models to better generate pseudo-labels and put them into a bootstrap framework for classifier training. More recently, label selection and debiasing have been studied too [3]. I believe some of these methods can be easily adapted as baselines here.
>
> Thank you for your baselines recommendations. Our goal was to compare our approach with SSL works that are similar (consistency regularization-based) and contemporary to our work. For completeness, we will add the suggested comparisons.

---

> > ### Comment · Reviewer_MfPv · 2024-12-02
> > **Thanks for the reply**
> >
> > On Cost-Effectiveness:
> > To evaluate cost-effectiveness, we should compare different methods under the same budget constraints. The currently proposed solution is likely not the optimal way to allocate resources. It's insufficient to simply argue that using a 7B model on a small dataset is inexpensive. We need a more constructive approach: given a fixed number of prompts, what are the most impactful examples to probe?
> >
> > On Baselines:
> > I believe weakly supervised methods can be seamlessly adapted to the SSL setting. Therefore, they should be compared systematically to ensure a fair evaluation.

---

### Official Review · Reviewer_hC3t · 2024-11-04

**Soundness:** 2
**Presentation:** 3
**Contribution:** 1
**Rating:** 3
**Confidence:** 5

**Summary:**

This paper introduces LLM-SSL, a method that combines FixMatch with LLMs for semi-supervised text classification. The authors claim that incorporating LLMs improves semi-supervised text classification performance compared to FixMatch and other SSL baselines.

**Strengths:**

The paper is clearly written and easy to follow.

**Weaknesses:**

1. Missing Important Baselines:
The paper omits several key baselines for semi-supervised text classification, such as PET [1], which also uses large amounts of unlabeled data for training. Additionally, it would be useful to evaluate the performance of using LLMs like Mistral-7B-Instruct directly for text classification without any fine-tuning. With few-shot examples, in-context learning could serve as an alternative for semi-supervised text classification. Recent work [2] also explores similar applications of LLMs in text classification and could provide additional comparative insights.

2. Contribution:
The use of LLMs for labeling is presented as a primary contribution of this paper. However, this approach is already well-established in NLP and may not suffice as a novel contribution.

3. Unfair Comparison:
The comparisons may not be entirely fair, as none of the baselines incorporate LLMs. Adding LLMs for labeling naturally boosts performance, making it difficult to isolate the effectiveness of the proposed SSL method. To present a fair comparison, it would be better to exclude LLMs. Without them, the performance of the proposed method appears to be close to that of other SSL baselines.

References:
[1] Schick et al., Exploiting Cloze Questions for Few-Shot Text Classification and Natural Language Inference
[2] Zhang et al., Generation-driven Contrastive Self-Training for Zero-Shot Text Classification with Instruction-Tuned GPT

**Questions:**

Why not use LLMs to estimate the quality of model-labeled examples?

---

> ### Author Response · Authors · 2024-11-28
>
> Thank you for your comments!
>
> > Missing Important Baselines: The paper omits several key baselines for semi-supervised text classification, such as PET [1], which also uses large amounts of unlabeled data for training. Recent work [2] also explores similar applications of LLMs in text classification and could provide additional comparative insights.
>
>
> Thank you for your baselines recommendations. Our goal was to compare our approach with SSL works that are similar (consistency regularization-based) and contemporary to our work. For completeness, we will add the suggested comparisons.
>
>
> > Additionally, it would be useful to evaluate the performance of using LLMs like Mistral-7B-Instruct directly for text classification without any fine-tuning. With few-shot examples, in-context learning could serve as an alternative for semi-supervised text classification.
>
>
> In L253-255 and L313 we note that we compare against the zero-shot Mistral-7B-Instruct model in our experiments. This experiment can be found in our main table as LLM-ZS. Our baseline description section (L313) points out that LLM-ZS represents the zero-shot LLM without any fine-tuning. Additionally, in L314 we also list the in-context learning approach and denote it by LLM-FS. Hence, we already have both these results in the paper.
>
> > Contribution: The use of LLMs for labeling is presented as a primary contribution of this paper. However, this approach is already well-established in NLP and may not suffice as a novel contribution.
>
> We would like to request the reviewer to share more information about this generic comment. Specifically, the reviewer hints that there are numerous studies that leverage LLMs as weak labelers in SSL frameworks. Could you provide some of these studies?
>
> > Unfair Comparison The comparisons may not be entirely fair, as none of the baselines incorporate LLMs. Adding LLMs for labeling naturally boosts performance, making it difficult to isolate the effectiveness of the proposed SSL method. To present a fair comparison, it would be better to exclude LLMs. Without them, the performance of the proposed method appears to be close to that of other SSL baselines.
>
> We remind the reviewer that our contribution is not only the AUM module, but also the teacher annealing technique that dynamically weighs the LLM prediction based on the learning status of the model. On another note, the statement that “Adding LLMs for labeling naturally boosts performance” is not necessarily True. For example, the LLM-DS (in the main table) approach, which leverages the labeled data and LLM pseudo-labeled unlabeled data during training lags behind in performance considerably compared to LLM-SSL. This experiment shows that naively using LLM labels does not boost the performance. One of our main contributions is identifying a way to utilize the LLM predictions more effectively. We achieve this by a teacher annealing framework where we weigh the LLM prediction higher if the base model has poor learning status and gradually lower the weight of LLM predictions (i.e., enable the small model to correct the incorrect LLM predictions) as the base model improves its learning status.

---

### Meta-Review · Area_Chair_Ltxj · 2024-12-19

**Metareview:**

This paper investigates semi-supervised text classification by leveraging large language models (LLMs). The proposed method utilizes LLMs to generate predictions on unlabeled examples and uses these predictions to guide the SSL training and improve the quality of pseudo-labels during training.  The experimental results show that the proposed method outperforms baseline methods, demonstrating that the integration of both methods is effective.

Pros:
- This paper is easy-to-follow.
- The motivation of the proposed method is intuitive.

Reasons to reject:
- Simply integrating LLMs into semi-supervised text classification is intuitive and somewhat straightforward, which makes the proposed method lack novelty, and thus the contributions are not significant enough.
- This paper did not compare with many other powerful semi-supervised learning methods. Although the authors argue that their goal is to compare the proposed method with semi-supervised learning methods that are similar (consistency regularization-based) and contemporary to their work. I do not think this is reasonable and fair.

**Additional Comments On Reviewer Discussion:**

This paper receives the scores of  5, 5, 5, 3, 1, which means that all the reviewers vote for rejecting this paper. The reviewers have raised many concerns, like the limited novelty/contributions of this work, the lack of comparison with related studies, and many other comments. For the comparison with other semi-supervised learning methods, the authors argue that their goal is to compare the proposed method with semi-supervised learning methods that are similar (consistency regularization-based) and contemporary to their work. I do not think this is reasonable and fair. The authors also indicated that they will add the suggested comparisons, but they did not provide any update on the experimental results of additional comparisons throughout the whole rebuttal period. Therefore, I agree with all the reviewers that the current version of this paper should be rejected.

---

### Decision · Program_Chairs · 2025-01-22

Reject